# Using Sandwiched Silicon/Reduced Graphene Oxide Composites with Dual Hybridization for Their Stable Lithium Storage Properties

**DOI:** 10.3390/molecules29102178

**Published:** 2024-05-07

**Authors:** Yuying Yang, Rui Zhang, Qiang Zhang, Liu Feng, Guangwu Wen, Lu-Chang Qin, Dong Wang

**Affiliations:** 1Analytical and Testing Center, Shandong University of Technology, Zibo 255000, China; yangyuying0533@163.com (Y.Y.); qzhang0533@163.com (Q.Z.); willow-feng@163.com (L.F.); 2School of Materials Science and Engineering, Shandong University of Technology, Zibo 255000, China; wengw@sdut.edu.cn; 3Department of Physics and Astronomy, University of North Carolina, Chapel Hill, NC 27599-3255, USA; 4Shangdong Si-Nano Materials Technology Co., Ltd., Zibo 255000, China

**Keywords:** silicon/reduced graphene oxide, duple combination, sandwiched structure, cycling performance

## Abstract

Using silicon/reduced graphene oxide (Si/rGO) composites as lithium-ion battery (LIB) anodes can effectively buffer the volumetric expansion and shrinkage of Si. Herein, we designed and prepared Si/rGO-b with a sandwiched structure, formed by a duple combination of ammonia-modified silicon (m-Si) nanoparticles (NP) with graphene oxide (GO). In the first composite process of m-Si and GO, a core–shell structure of primal Si/rGO-b (p-Si/rGO-b) was formed. The amino groups on the m-Si surface can not only hybridize with the GO surface to fix the Si particles, but also form covalent chemical bonds with the remaining carboxyl groups of rGO to enhance the stability of the composite. During the electrochemical reaction, the oxygen on the m-Si surface reacts with lithium ions (Li^+^) to form Li_2_O, which is a component of the solid–electrolyte interphase (SEI) and is beneficial to buffering the volume expansion of Si. Then, the p-Si/rGO-b recombines with GO again to finally form a sandwiched structure of Si/rGO-b. Covalent chemical bonds are formed between the rGO layers to tightly fix the p-Si/rGO-b, and the conductive network formed by the reintroduced rGO improves the conductivity of the Si/rGO-b composite. When used as an electrode, the Si/rGO-b composite exhibits excellent cycling performance (operated stably for more than 800 cycles at a high-capacity retention rate of 82.4%) and a superior rate capability (300 mA h/g at 5 A/g). After cycling, tiny cracks formed in some areas of the electrode surface, with an expansion rate of only 27.4%. The duple combination of rGO and the unique sandwiched structure presented here demonstrate great effectiveness in improving the electrochemical performance of alloy-type anodes.

## 1. Introduction

The storage and utilization of electrochemical energy is arousing a great deal of interest because of the numerous practical applications of these technologies, such as batteries for electric vehicles and portable electronic devices [1,2]. Rechargeable lithium-ion batteries (LIBs) have become favored devices largely due to their attractive advantages regarding their high energy density [3]. On the other hand, compared with traditional graphite electrodes (with a theoretical capacity of 372 mA h/g), Si has become widely regarded as a more promising negative electrode material for LIBs because of its high theoretical capacity (ca. 4200 mAh/g for Li_4.4_Si) and low working potential (<0.5 V, versus Li^+^/Li) [4,5]. Unfortunately, the volumetric expansion of Si during the lithiation and delithiation processes is high, reaching up to 300% [6], leading to rapid collapses of the electrode. Many technical strategies have been explored in an attempt to overcome the challenges of the volumetric expansion of Si electrodes, including (a) reducing the Si particle size by preparing nanoscale Si structures [7,8], (b) optimizing the morphology and structure of silicon material, for example, the Si nanowires [9], Si nanotubes [10], and/or porous Si [11,12,13,14], (c) utilizing new binders to improve the stability of the Si electrode [15], (d) compositing with other materials and so on. Among these, by compositing with other materials [16,17,18], composite Si electrodes have also often been applied to improve the electrochemical performance of LIBs [19].

Carbon-based Si composites have shown both great mechanical flexibility and good conductivity to tackle the problems of volumetric expansions and poor conductivity of the Si anode [20,21]. Such composites have been made with glucose [14], chitosan [22], resin [23], dopamine [24,25], and organic polymers [26,27]. Graphene is a two-dimensional material with outstanding electrical conductivity and mechanical properties that can be used to buffer the volumetric expansion of the Si electrode [2]. Although graphene–silicon composites have exhibited good cycle performance and rate capability as anodes of LIBs, graphene has low reactivity and it often requires complex processing procedures to form a composite with Si with an encapsulated structure [28]. As a result, graphene–silicon composites are still limited in their practical applications.

Graphene oxide (GO), as a precursor for producing rGO, contains different functional groups and can easily be dispersed to form a suspension with a low Zeta potential [29,30,31], which is in turn helpful for it to self-assemble easily and quickly with Si to form a Si/GO composite. A Si/rGO composite can then be obtained easily from Si/GO when GO is reduced [32,33]. Therefore, many Si/rGO composites have been designed and applied as negative electrode materials for lithium-ion batteries [34,35]. However, the rapidly expanding volume of Si particles leads to a decrease in the conductivity of the active material and partial detachment from the collector during the lithiation process. That means that more stable attachment and an unhindered conductive network are urgent needed for Si/carbon composites, which is also a challenge faced by Si/rGO negative electrode materials.

Herein, we report a facile method for preparing Si/rGO-b composites. Firstly, ammonium hydroxide is used to modify the surface of Si NPs to form (NH_4_)_2_SiO_3_. Amino groups enable Si particles to be positively charged and self-assemble with GO to form the primal Si/rGO-b (p-Si/rGO-b) composites. Secondly, the primal Si/rGO-b is further covered by GO again, and then is reduced to form the final product of Si/rGO-b. In this composite, the m-Si particles are first tightly wrapped by rGO to form a core–shell structure. The amino groups on the m-Si surface can not only hybridize to the rGO surface to fix the Si particles, but also form covalent chemical bonds with the remaining carboxyl groups of rGO to enhance the stability of the composite. During the electrochemical reaction, the oxygen on the m-Si surface and lithium ions (Li^+^) form Li_2_O, which, as a component of the solid–electrolyte interphase (SEI), can improve the stability of the SEI film. The SEI film can assist rGO in buffering the volume expansion of Si. The core–shell structure of p-Si/rGO-b is further covered by rGO to form a sandwiched structure of Si/rGO-b. At the same time, more covalent chemical bonds are formed between the reintroduced rGO layers to tightly fix the p-Si/rGO-b, and the conductive grid formed by the reintroduced rGO improves the conductivity of the Si/rGO-b composite. As a result, the Si/rGO-b composite electrode presented here shows excellent electrochemical performance.

## 2. Results and Discussion

### 2.1. Structural Characterization

The contrast sample of Si/rGO-a was self-assembled following the route illustrated in Appendix A, with the same amount of GO but only wrapped once. The synthetic process of the Si/rGO-b composite is depicted schematically in Figure 1a. Firstly, Si NPs were dispersed in a mixture of ethanol and water and then NH_3_·H_2_O was applied to modify the Si surfaces. NH_3_·H_2_O reacted with Si to form (NH_4_)_2_SiO_3_ [36,37] (Appendix A), making the surface of the Si NPs rich in amino groups. The amino-rich Si NPs and GO self-assembled to form the p-Si/GO composite, which was then reduced with sodium citrate to obtain the primary p-Si/rGO-b composite. After mechanical grinding and ultrasonic dispersion, the p-Si/rGO-b composite was modified by NH_3_·H_2_O, and then was composited with GO again and a sandwiched structure of Si/rGO-b composite was obtained after being reduced with N_2_H_4_·H_2_O. Through NH_3_·H_2_O, the C-N bonds were formed on the rGO basal plane and the C-O-N bonds (Appendix A) which formed by carboxyl and amino groups between the rGO layers helped to stabilize the composite structure. According to atomic force microscopy (AFM) analysis (Appendix A), the thickness of the GO was about 0.8 nm, representing a single layer of graphene oxide. A single layer of GO has the largest specific surface area and can better disperse Si NPs. The SEM images (Figure 1b,c) illustrate that Si/rGO-b and Si/rGO-a have different morphologies. In Figure 1b, the p-Si/rGO-b particles formed by the first combination of GO and Si exhibit a core–shell structure. Then, the reintroduced GO connected p-Si/rGO-b particles to the rGO layer, forming the sandwiched structure of Si/rGO-b. The morphology of Si/rGO-a formed from GO and Si is a simple core–shell structure, similar to that of p-Si/rGO-b.

The microstructure of Si, m-Si and the Si/rGO-b composite was examined via transmission electron microscopy (TEM). Figure 2a shows a TEM image of typical Si NPs with a particle size of 50–100 nm. The corresponding high-resolution TEM (HRTEM) image is shown in Figure 2d. It can be observed that there was a thin amorphous layer on the surface of Si. As revealed in Figure 2b, after modification with NH_3_·H_2_O, the morphology of the Si NPs was not changed significantly, but the thickness of the amorphous film (Labeled Si-O) on the Si surface increased (Figure 2e). This indicates that NH_3_·H_2_O undergoes a chemical reaction with Si NPs, resulting in the formation of amorphous structures on the Si surface and the attachment of amino groups. Figure 2c is a TEM image of the Si/rGO-b. The Si NPs were uniformly dispersed on the surface of rGO or covered with rGO (Figure 2f). Figure 2g is a STEM image of the Si/rGO-b composite. Figure 2h–j present the elemental mappings of the Si/rGO-b composite, revealing a homogeneous distribution of Si (Figure 2h), N (Figure 2i), and O (Figure 2j) in the composite. The uniform distribution of elements indicates that the formed chemical bonds, C-N and C-O-N, were also present everywhere in the composite, making the sandwiched structure more stable.

A full-survey X-ray photoelectron spectroscopy (XPS) spectrum of the Si/rGO-b composite is shown in Appendix A, from which we can find that the main elements were C and Si, while some oxygen and a small amount of nitrogen also remained in Si/rGO. The presence of oxygen was due to the incomplete reduction of GO and oxides on the Si surface. The nitrogen came from NH_3_·H_2_O and N_2_H_4_·H_2_O. Figure 3a is the 2p orbital spectrum of Si, composed of two peaks. The main two peaks are located at 99.6 eV and 103.7 eV, corresponding to the Si-Si and Si-O bond, respectively [5,38]. Figure 3b is the C 1s orbital spectrum, in which the C-N single bond is observed clearly, indicating that the nitrogen atoms were doped onto the GO basal surface during the reduction of GO. These nitrogen atoms can form a stable C-O-N bond with the carboxyl groups, establishing connection brackets on different rGO basal surfaces to make the sandwiched structure more stable.

The Raman spectra of rGO and Si/rGO-b from 300 to 2300 cm^−1^ are displayed in Figure 3c. The two typical peaks are located at about 1350 cm^−1^ and 1600 cm^−1^, corresponding to the disordered (D band) and graphitic structure (G band) of rGO, respectively [39,40]. In the spectrum of the Si/rGO composite, the characteristic peaks of Si can be detected, including the first-order Raman scattering peak of Si-Si bond stretching vibrations with a sharp peak at 515 cm^−1^ and a broad peak at 950 cm^−1^ [41,42]. This shows that Si maintained its crystallinity during the reactions. In the Si/rGO-b composite, the intensity ratio of the D peak to the G peak is 1.3, which is similar to the ratio of rGO. These pore channels can promote the permeation of the electrolyte into the electrode and provide faster diffusion channels for lithium ions (Li^+^) [43]. The diffractogram of XRD for Si/rGO-b and Si/rGO-a shows the characteristic peaks of Si (111), (220) and (311), indicating that, after synthesis, the crystalline Si does not change (Figure 3b). The amorphous peak at 2θ = 24.5° is caused by amorphous silicon on the Si surface, indicating a chemical reaction between ammonia and Si. The specific surface area of Si/rGO-b, Si/rGO-a and Si is 183.8, 119.7, and 17.1 m^2^/g, respectively, as shown in Figure 3e. Benefiting from the introduction of rGO with a large specific surface area, the specific surface area of Si/rGO-b and Si/rGO-a is much higher than that of pure Si. The higher specific surface area of Si/rGO-b than Si/rGO-a may be due to the repeated recombination of Si and GO. At the same time, the isotherm of Si/rGO-b exhibits a typical hysteresis loop, indicating a porous structure. On the other hand, the adsorption and desorption isotherm of pure Si shows no hysteresis loop, indicating that it has a non-porous structure. The average pore diameter of Si/rGO-b is 4.0 nm, (Figure 3e), which is very conducive to the transmission of Li^+^. Figure 3f shows that the electrical conductivity of Si/rGO-b, Si/rGO-a and Si is 0.0386, 0.0283 and 0.00053 S/mm, respectively. The good conductivity of the rGO network significantly improves the conductivity of the composite material, Si/rGO-b.

Appendix A shows the thermogravimetric (TG) decomposition curve of the Si/rGO-b and Si/rGO-a composites and the oxidation curve of pure Si in air from 30 to 800 °C. The decomposition temperature of Si/rGO-b is higher than that of Si/rGO-a, indicating that the sandwiched structure of Si/rGO-b makes it more stable than Si/rGO-a. Using HF to treat the composites to exclude the Si-O, and then NaOH to treat the composites exclude the Si, the proportions of Si, Si-O, and rGO in the Si/rGO-b and Si/rGO-a composites were calculated (Appendix A). The proportions of rGO, Si-O and Si in Si/rGO-b determined by the subtraction method are 42.7%, 7.8%, and 49.5%, respectively. The corresponding component contents in Si/rGO-a are 45.1%, 8.0% and 46.9%, respectively. The component content of the two composite materials is similar.

### 2.2. Electrochemical Characterization

In the cyclic voltammetry (CV) curves as shown in Figure 4a, a broad cathodic peak of low intensity appears in 0.1–0.5 V in the first circle, indicating that an SEI film forms on the surface of the electrode [44,45]. The formation of the SEI film is an irreversible process and consumes amount of Li^+^, resulting in low CE during the first charge and discharge cycle of the battery. When the cathodic peak is lower than 0.1 V, the Si NPs undergo a lithiation reaction and the crystalline Si NPs are converted to amorphous Li_x_Si due to the intercalation of Li^+^ [41]. In the following anodic scan, two peaks at 0.28 V and 0.5 V appear, caused by the decomposition of amorphous Li_x_Si at low and high voltages, respectively [46]. After the first cycle, because of the insertion of lithium and the removal of Si, the activity of the electrode is enhanced. At the same time, the currents of the two oxidation peaks also increase gradually. Because of the lithiation reaction of amorphous Si, a strong reduction peak appears at 0.23 V from the second circle [47].

Appendix A show the initial discharge/charge voltage curve of the Si/rGO-b, Si/rGO-a and Si. Appendix A present the corresponding second to fifth discharge/charge voltage curves. The initial coulombic efficiency (ICE) of Si/rGO-b, Si/rGO-a and Si are 29.05%, 56.78% and 75.77%, respectively. The initial discharge curves of Si/rGO-b and Si/rGO-a show that the discharge platforms below 0.1 V are very short, indicating that the irreversible reaction of SEI film formation consumes a large amount of Li^+^ and results in lower ICE [48,49]. These irreversible reactions include the reaction of Li^+^ with the electrolyte [50]. When the current density is high, Li^+^ also reacts with amorphous oxygen on the surface of Si to form Li_2_O and Li_2_SiO_4_ [51,52]. Figure 4b presents the discharge/charge voltage curves of the Si/rGO-b composite at various current densities and shows the obvious potential plateaus. As the current density increases, the specific capacity of the battery gradually decreases. When the current density is 2 C, the specific capacity of the battery is 450 mA h/g, which is much higher than the theoretical specific capacity of graphite (372 mA h/g). When the current density is 5 C, the Si/rGO-b electrode still exhibits a high capacity of 334 mA h/g.

**Figure 4 molecules-29-02178-f004:**
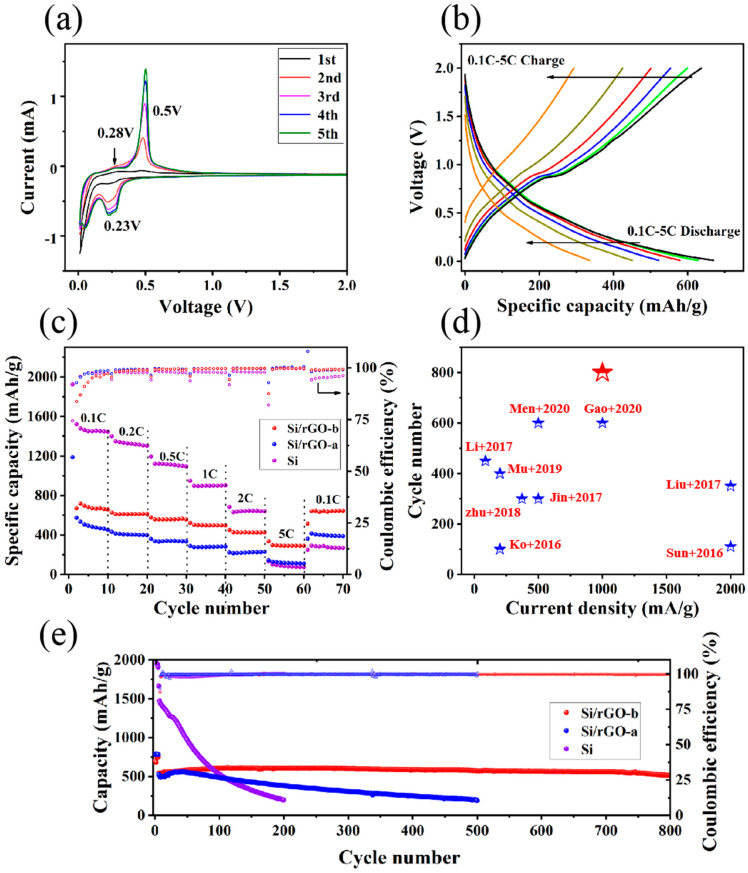
Electrochemical characterization of electrodes. (**a**) CV curves of the Si/rGO-b composite at a scan rate of 0.1 mV/s within the potential range of 0.01–2.0 V. (**b**) Charge/discharge curves at different current densities of Si/rGO-b (0.1 C–5 C, 1 C = 1 A/g). (**c**) Rate performance of Si/rGO-b, Si/rGO-a and pure Si. (**d**) Comparison of lithium storage properties with previous work on Si-based anodes ([5,25,53,54,55,56,57,58,59], our work: red star). (**e**) Capacity and CE of Si/rGO-b (800 cycles), Si/rGO-a (500 cycles) and Si (200 cycles) at 1 C.

To evaluate the rate performance of the Si/rGO-b composite electrode in comparison with Si/rGO-a and Si, these electrodes were subjected to charge and discharge cycles with a current density from 0.1 to 5 C, and the results are shown in Figure 4c. As the current density increases, the specific capacity of the Si/rGO-b composite electrode decreases slowly from about 670 to 300 mA h/g, while that of the Si/rGO-a electrode decreases much more rapidly from 574 to 143 mA h/g, and the specific capacity of the Si electrode decreases even more rapidly from 1700 to 102 mA h/g. At a current density of 5 C, the specific capacity of the Si/rGO-b (300 mA h/g) is much higher than that of Si/rGO-a (143 mA h/g) and Si (102 mA h/g). Furthermore, after the current density returns to 0.1 C, Si/rGO-b shows a superior reversible specific capacity to 645 mA h/g, with 96.3% of the remaining initial specific capacity (670 mA h/g), compared to that for Si/rGO-a and Si (400 mA h/g and 69.7% for Si/rGO-a and 285 mA h/g and 6.0% for Si).

Cycling stability is one of the most important performance indicators of a battery. As shown in Figure 4e, after being activated with a current density of 0.1 C, the half-cell can be charged and discharged stably at a current density of 1 C for 800 cycles. The Si/rGO-b composite gives rise to a reversible capacity of 620 mA h/g in the initial cycle and displays a specific capacity retention ratio of 82.4%, reducing to 511 mA h/g after 800 cycles. The initial specific capacity of Si/rGO-a is 528 mA h/g, and after 500 cycles, the specific capacity decays to 200 mA h/g. As a comparison, the cycle stability of pure Si drops much more rapidly. After 200 cycles, the specific capacity of the pure Si electrode decays from 1662 to 198 mA h/g. This outstanding stability for Si/rGO-b could be attributed to the following factors: Firstly, the core–shell structure (p-Si/rGO-b) formed by the first recombination of rGO and Si partially buffers the expansion of Si. Secondly, the oxygen on the surface of m-Si forms Li_2_O with Li^+^, enhancing the stability of the SEI film. The stable SEI film can assist rGO in buffering the volume expansion of Si. Thirdly, compared to the Si/rGO-a composite, the sandwiched structure of Si/rGO-b formed by reintroducing rGO and p-Si/rGO-b exhibits better stability and conductivity. These three reasons cause the Si/rGO-b composite to have better cycling performance than Si/rGO-a and Si. As illustrated in Appendix A and Figure 4d, this Si/rGO-b composite electrode exhibited performance superior to those reported in the literature.

Appendix A and Figure 5a show the Nyquist plots of the EIS data for the Si/rGO-b, Si/rGO-a and Si electrodes before and after 50 cycles. Among the simulated kinetic parameters, *R_s_* represents the electrolyte resistance, which is mainly caused by the electrolyte, current collector, and electrode material [60]. Its value is the intercept of the curve on the horizontal axis at a high frequency. Appendix A shows that the initial *R_s_* of Si/rGO-b, Si/rGO-a and Si electrodes is 4.39, 5.04, and 8.46 Ω, respectively, and after 50 cycles, the corresponding *R_s_* is 1.95, 2.23 and 4.99 Ω, respectively. At the high- to middle-frequency region, the semicircle represents the charge-transfer resistance (*R_ct_*) related to the active material. In Figure 5b, the initial values of *R_ct_* of the Si/rGO-b and Si/rGO-a composites and pure Si are 136.9, 177.6 and 371.2 Ω, respectively, and after 50 cycles, the corresponding *R_ct_* are 51.7 Ω for Si/rGO-b, 57.3 Ω for Si/rGO-a and 189.1 Ω for Si. The values of *R_s_* and *R_ct_* for the Si/rGO-b and Si/rGO-a composites are smaller than those of pure Si, mainly because the introduced graphene has excellent electrical conductivity. Furthermore, the sandwiched structure of Si/rGO-b provides it with excellent conductivity, so its *R_s_* and *R_ct_* values are smaller than those of Si/rGO-a.

The diffusion rate of Li^+^ is an important factor affecting the cycling performance of batteries [32]. In the equivalent circuit of EIS curve, CPE1 represents the double layer capacitance. In the low frequency of EIS curve, *W* is the Warburg impedance for the diffusion of Li^+^ in the electrode (Figure 5a). According to Formulas (1) and (2) [61],
(1)$$Z_{real}=R_{e}+R_{ct}+\sigma \omega^{-1/2}$$
(2)$$D=\frac{R^{2}T^{2}}{2A^{2}n^{4}F^{4}C^{2}\sigma^{2}},$$
the Li^+^ diffusion rate of Si/rGO-b, Si/rGO-a, and Si electrodes can be calculated. In Formula (1), *Z_real_* is the resistance of the electrolyte on the real axis, *R_e_* is the resistance between the electrolyte and electrode, *R_ct_* is the charge transfer resistance, and *w* is the angle frequency. In Formula (2), *R* represents the gas constant, *T* is the absolute temperature, *A* is the area of the electrode disk, *F* is the Faraday constant, and *n* represents the number of electron transfers per molecular reaction. According to Formula (1), the slope *σ* of Si/rGO-b, Si/rGO-a and Si is 27.2, 41.9 and 58.6 (Figure 5c inset), respectively. According to Formula (2), the *D*_Li+_ values calculated using the *σ* value are 3.72 × 10^−14^ cm^2^/s for Si/rGO-b, 1.57 × 10^−14^ cm^2^/s for Si/rGO-a and 8.02 × 10^−15^ cm^2^/s for Si, respectively (Figure 5c). The large value of *D*_Li+_ for Si/rGO-b indicates that the sandwiched structure with the primal core–shell structure (p-Si/rGO-b) formed by the graphene which was introduced again is conducive to the transport of Li^+^.

To further investigate the Li^+^ storage properties of the Si/rGO-b electrode, CV tests were conducted at different scan rates to study the reaction kinetics, as shown in Figure 5d. In Formula (3) [62], the peak current (*i*) and scan rate (*v*) follow a power-law relationship. When *b* = 0.5, the electrochemical behavior is mainly controlled by a solid-state process; when *b* = 1.0, the capacity is dominated by surface-controlled reactions. In Figure 5e, the *b* value is 0.73 according to the relationship between log^(*i*)^ and log^(*v*)^. The *b* value indicates that the electrochemical behavior of Si/rGO-b is controlled by both a solid-state process and surface-controlled reactions. In Formula (4) [63], *k*_1_*v* represents the capacitive contribution and *k*_2_*v*^1/2^ represents the diffusion-controlled contribution. In Figure 5f, the scan rates increase from 0.2 to 1.0 mV/s, and the respective fraction of the capacitive contribution increases from 57% to 95%.
*i* = *av^b^*(3)
*i* = *k*_1_*v* + *k*_2_*v*^1/2^(4)

To further investigate the structural stability of the Si/rGO-b, Si/rGO-a and Si electrodes, the morphology of the electrodes after 200 charge and discharge cycles was observed with SEM. For the Si/rGO-b electrode, as shown in Figure 6a, only tiny cracks at fewer locations were observed. Meanwhile, he Si/rGO-a (Figure 6b) electrode displays noticeable cracks, and the Si (Figure 6c) electrode undergoes pulverization and severe cracks appear on the entire surface. Figure 6c–k show the longitudinal expansion of the electrode material. The Si/rGO-b (Figure 6d,g) composite electrode expands from 19.7 μm to 25.1 μm, with an expansion rate of 27.4%. The Si/rGO-a (Figure 6e,h) composite and Si (Figure 6f,i) electrode expand from 20.2 μm to 28.7 μm, with an expansion rate of 42%, and from 16.9 μm to 37.4 μm, with an expansion rate of 121%, respectively.

The morphological changes of the active materials Si/rGO-b, Si/rGO-a, and Si during lithiation/delithiation processes are illustrated in Figure 7. As an electrode material, Si forms Li_x_Si during the lithiation process, and the volume of Li_x_Si rapidly increases. After delithiation, Li_x_Si forms amorphous Si, and Si undergoes pulverization. The contact level between small-sized Si particles is poor, and some Si falls off the copper collector, leading to a rapid decline in electrochemical performance. Due to the buffering effect of rGO, the volume expansion of Si/rGO-a is relatively small. In the delithiation reaction, the amorphous Si is confined in the core structure with the tight encapsulation of Si particles by rGO. However, the expansion/shrinkage of the Si particles leads to a poor contact level between the Si/rGO-a composite, resulting in a decrease in the conductivity of Si/rGO-a, and a small amount of Si peels off from the rGO coating layer to form bare Si, resulting in a slow decline in electrochemical performance. For Si/rGO-b, the core–shell structure with the tight encapsulation of rGO can restrict the volume expansion/shrinkage and prevent Si pulverization. Moreover, the second introduction of rGO to form a sandwiched structure further guarantees the continuous conductivity network and structural integrity, which induces good electrochemical performance. Therefore, this superior cycling stability is attributed to the following merits: (i) In the core–shell structure of primal Si/rGO-b (p-Si/rGO-b), the good mechanical properties of rGO partially buffer the expansion of Si. Covalent chemical bonds fix the Si and enhance the stability of p-Si/rGO-b. (ii) The oxygen on the surface of m-Si reacts with Li^+^ to form Li_2_O. As a component of the SEI film, Li_2_O enhances the stability of the SEI film, and the stable SEI film can assist rGO in buffering the volume expansion of Si. (iii) The design of the sandwiched structure of Si/rGO-b results in better stability and conductivity than Si and the Si/rGO-a composite.

## 3. Materials and Methods

Synthesis of GO: Information on the purchase of materials is shown in Appendix A. GO was synthesized by an improved Hummers method [31]. Concentrated H_2_SO_4_ (120 mL) was put into a flask (500 mL), under mechanical stirring (200–250 rpm), graphite powder (3.0 g), NaNO_3_ (2.0 g), and KMnO_4_ (12.0 g) were slowly put into this flask and kept the temperature of the suspension maintained at temperature of 20 ± 2 °C and stirred (200–250 rpm) for about 1 h to obtain GO. Then, the obtained GO was washed with HCl and deionized water to obtain a neutral liquid. Finally, the obtained GO was dried in a freeze dryer and dispersed ultrasonically in deionized water to form a suspension with a concentration of 1.0 mg/mL.

Synthesis of the Si/rGO-b composite: In this process, 200 mg of Si NPs and 100 mL of ethanol were mixed and dispersed ultrasonically for 2 h. Then, 50 mL of deionized water and 2 mL of ammonium hydroxide (NH_3_·H_2_O) were added into the reactor and treated ultrasonically for 0.5 h. As a result, the surface of Si NPs was modified by NH_3_·H_2_O and the material was labeled as m-Si. m-Si was then mixed in 100 mL of GO solution (1 mg/mL) and magnetically stirred for 6 h to obtain homogenous p-Si/GO-b. m-Si has a positive Zeta potential, whereas the Zeta potential of GO is negative. Through electrostatic interaction, m-Si was dispersed on the GO sheets to realize self-encapsulation. Subsequently, sodium citrate (0.5 g) was added (reacted for 12 h in water bath at 80 °C) and the obtained material p-Si/rGO-b was washed with deionized water and dried in vacuum at 60 °C.

The dried p-Si/rGO-b was ground in a mortar before a mixture of ethanol and deionized water was added to disperse the suspension ultrasonically for 0.5 h. Subsequently, 2 mL of ammonium hydroxide and 150 mL (1 mg/mL) of GO solution were added and stirred magnetically for 6 h, and a Si/rGO/GO-b composite was formed. By adding 150 μL of N_2_H_4_·H_2_O, reactions were carried out for 12 h in a water bath of 85 °C. The product was then filtered with an organic filter membrane, washed three times with deionized water, and dried in vacuum for 10 h at 60 °C to obtain the Si/rGO-b composite.

Synthesis of Si/rGO-a composite: According to the schematic in Appendix A, 200 mg of m-Si was combined with 250 mL of GO (1 mg/mL) to form Si/GO, which was then reduced by N_2_H_4_·H_2_O hydrazine hydrate to obtain the Si/rGO-a composite.

Electrochemical Characterization: The active materials (Si/rGO-b, Si/rGO-a, Si), acetylene black, and sodium alginate (binder) were mixed in deionized water (2 mL) at a mass ratio of 8:1:1, and the mixture was stirred magnetically for 24 h to make sure that the materials were fully mixed. The resulting mixture was coated onto a copper foil and dried in vacuum at 80 °C for 3 h. Then, the final copper foil was cut into 14 mm disk electrodes. The 14 mm disk electrodes were further dried at 80 °C in vacuum for 12 h. The half-cell was assembled using a CR2025 battery case with a lithium foil as the anode and a Klud 2325 membrane as the separator. The electrolyte was LiPF_6_ (1 mol/L), composed of ethylene carbonate (EC) and diethyl carbonate (DEC) with a volumetric ratio of 1:1, and it contained 5% fluoroethylene carbonate (FEC). After it was assembled, the half-cell was shelved for 24 h before electrochemical measurements. Cyclic voltammetry measurement was performed at a scan rate of 0.1 mV/s and electrochemical impedance spectroscopy (EIS) measurement was performed within a frequency range of 0.01 Hz–100 kHz.

## 4. Conclusions

We have prepared a Si/rGO-b composite material with a sandwiched configuration by means of the repeated combination of ammonia-modified silicon (m-Si) and graphene oxide (GO). When used as an electrode material for LIBs, the Si/rGO-b composite exhibited excellent rate performance and cycle stability; after 800 cycles at 1 A/g, it retained 82.4% of its specific capacity, reaching up to 511 mAh/g. Charging and discharging at a current density of 0.1–5 A/g, the remaining specific capacity reaching 96.3% of the initial specific capacity. The superior electrochemical performance of the Si/rGO-b composite is ascribed to this unique sandwiched structure. Firstly, the core–shell-structure primal Si/rGO-b (p-Si/rGO-b) which forms in the first composite process not only fixes the Si NPs but also partially buffers the volume expansion of Si. Secondly, during the electrochemical process, the oxygen on the surface of m-Si forms Li_2_O with Li^+^ to enhance the stability of the SEI film, which can assist rGO in buffering the volume expansion of Si. Finally, the sandwiched structure with covalent bonds enhances the stability and conductivity of Si/rGO-b.

## Figures and Tables

**Figure 1 molecules-29-02178-f001:**
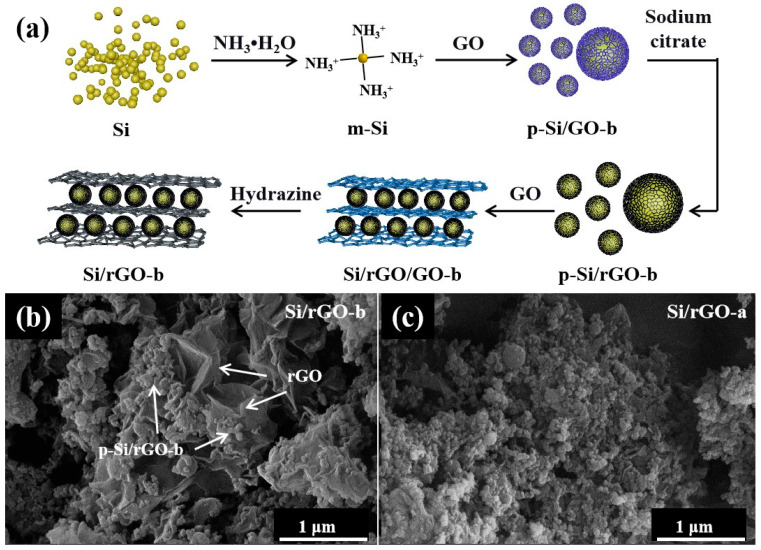
(**a**) Schematic illustration of the preparation of the Si/rGO-b composite. (**b**,**c**) SEM images of the Si/rGO-a and Si/rGO-b composites.

**Figure 2 molecules-29-02178-f002:**
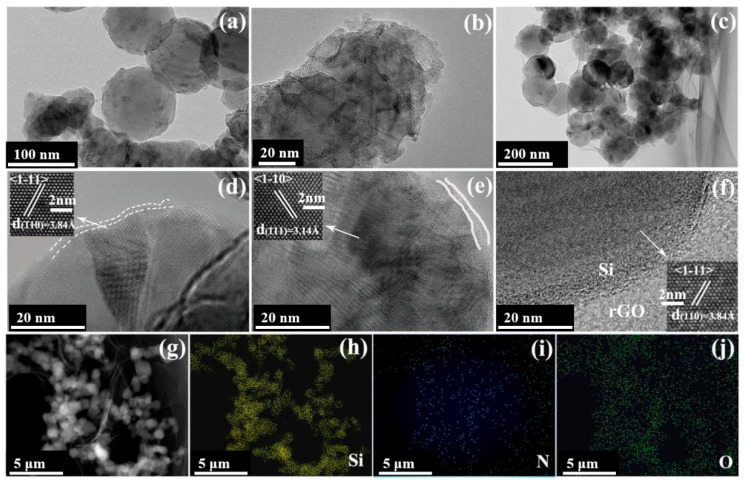
(**a**–**c**) TEM images of Si, m-Si, and the Si/rGO-b composite. (**d**–**f**) HRTEM images of Si, m-Si, and the Si/rGO-b composite. Insets are magnified HRTEM images of the marked regions. (**g**) STEM image and (**h**–**j**) EDS mapping of the Si/rGO-b composite.

**Figure 3 molecules-29-02178-f003:**
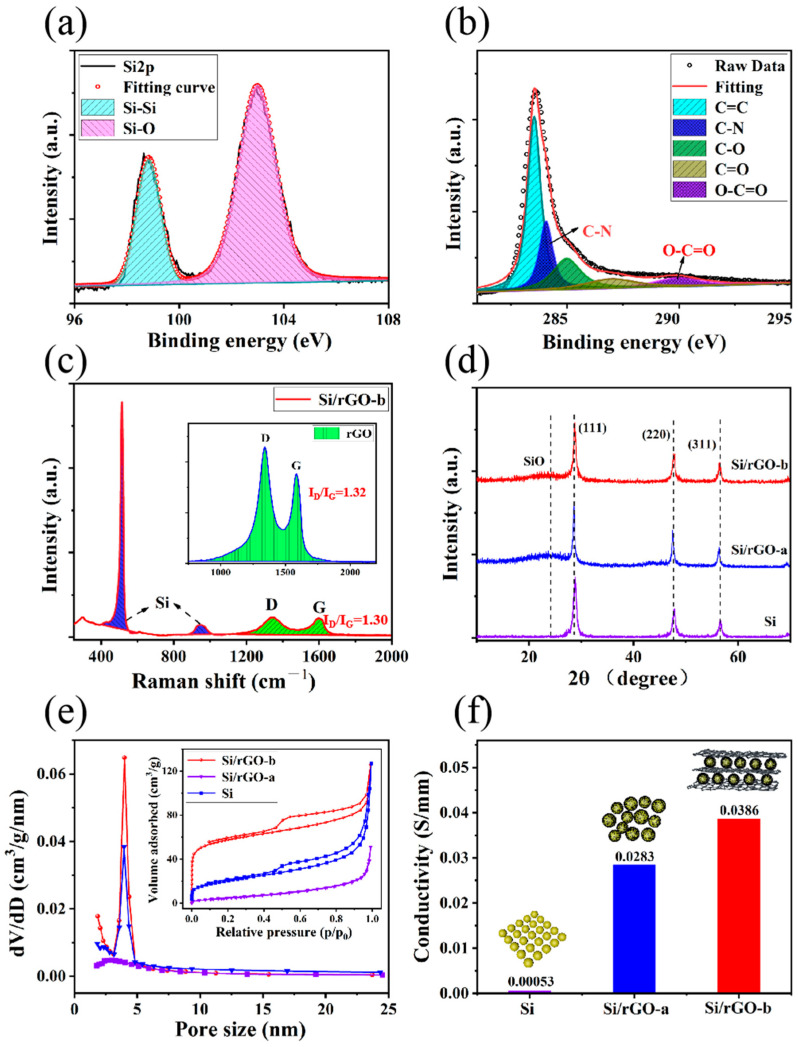
(**a**,**b**) XPS data of Si/rGO-b Si2p and C1s, respectively. (**c**) Raman spectra of rGO and Si/rGO-b. (**d**) XRD of Si, Si/rGO-a, and Si/rGO-b. (**e**) The pore size distribution of Si, Si/rGO-a and Si/rGO-b. Inset is the nitrogen adsorption/desorption isotherms of Si, Si/rGO-a and Si/rGO-b. (**f**) Conductivity of Si, Si/rGO-a and Si/rGO-b.

**Figure 5 molecules-29-02178-f005:**
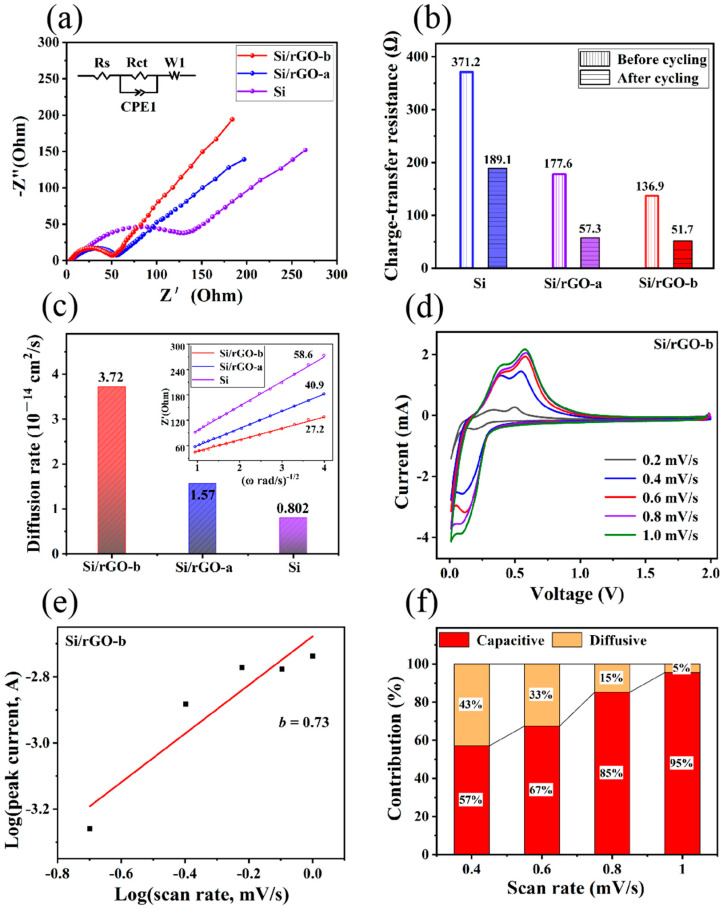
(**a**) Nyquist plots of EIS data for the Si/rGO-b, Si/rGO-a and Si electrodes after 50 discharge/charge cycles. Inset shows the equivalent circuit for the electrodes. (**b**) *R_ct_* of Si/rGO-b, Si/rGO-a and Si electrodes before and after 50 cycles. (**c**) Lithium ion diffusion rate of Si/rGO-b, Si/rGO-a and Si. Inset is the corresponding slope *σ* value of three types of electrodes. (**d**) CV curves at different scan rates from 0.2 to 1.0 mV/s of the Si/rGO-b electrode. (**e**) *b* value of the Si/rGO electrode via the relationship of log(*i*) vs. log(*v*). (**f**) Ratio of the capacitive contribution at different scan rates.

**Figure 6 molecules-29-02178-f006:**
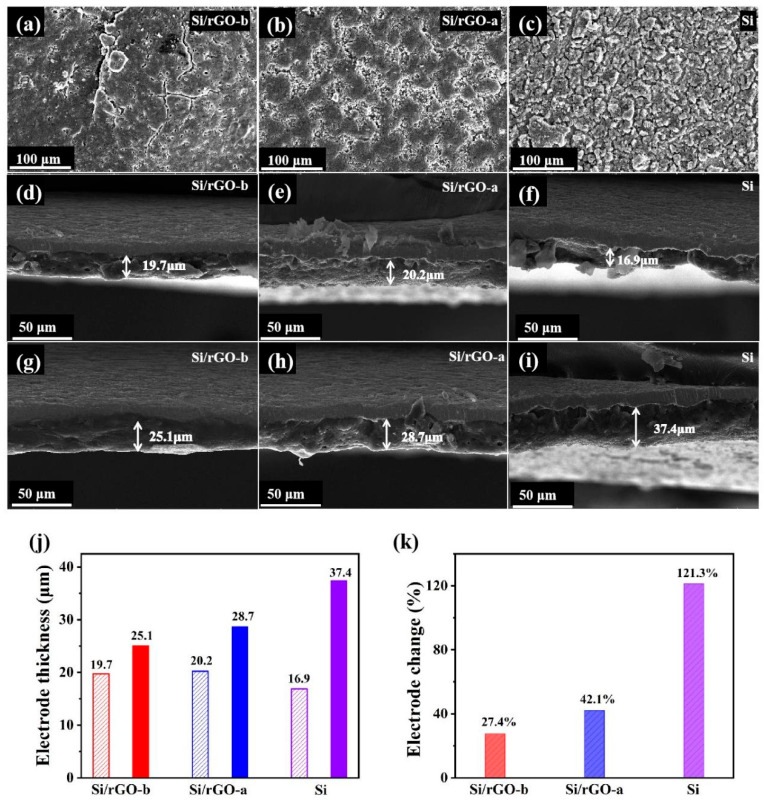
(**a**–**c**) SEM images of the (**a**) Si/rGO-b, (**b**) Si/rGO-a and (**c**) Si electrodes before and after 200 cycles, respectively. (**d**–**i**) Cross-sectional view of the Si/rGO-b (**d**,**g**), Si/rGO-a (**e**,**h**) and (**f**,**i**) Si electrodes before and after 200 cycles, respectively. (**j**) The thickness variation of Si/rGO-b, Si/rGO-a and Si electrode before and after 200 cycles. (**k**) The corresponding percentage change in the thickness of the three types of electrodes.

**Figure 7 molecules-29-02178-f007:**
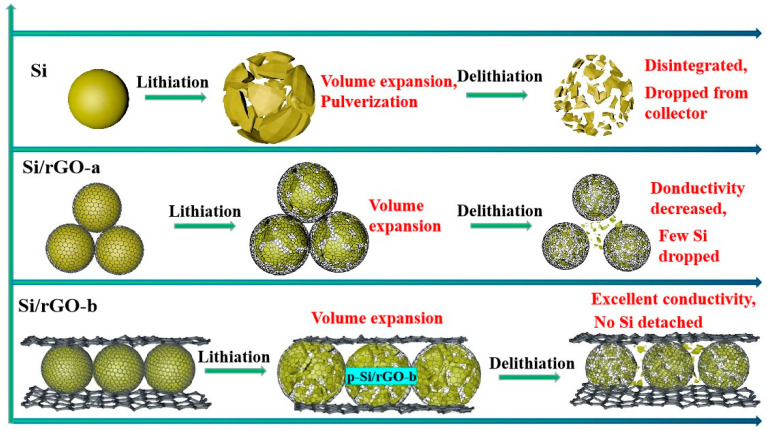
Morphological changes of the active materials Si, Si/rGO-a and Si/rGO-b during lithiation/delithiation processes.

## Data Availability

All data generated or analyzed during this study are included in this article.

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
