# Peer review of "Using Sandwiched Silicon/Reduced Graphene Oxide Composites with Dual Hybridization for Their Stable Lithium Storage Properties"

_molecules, 2024, doi:10.3390/molecules29102178_

Round 1

Reviewer 1 Report

Comments and Suggestions for Authors

The development of high-capacity anode materials is of critical importance to lithium-ion batteries. This work prepared a silicon/rGO composite and exhibited an excellent battery performance, including a high capacity, a high rate, and long cycling life. A set of characterization methods have been employed to interpret the research results. Overall, the paper is well prepared and organized, and I suggest publication after some minor changes.

1. The authors mentioned Figure S2 and Figure S3a earlier than Figure S1. Please revise their order.

2. In Figure 4d, the authors need to add “this work” inside the figure.  

3. In Figure 5e, the linear relationship between Log(current) and Log(scanning rate) is poor. Usually, most literature can have an excellent linear result with a R2 = 0.99-1. However, this work did not have good linearity, which makes the b value and capacitive/diffusive capacity contribution questionable and problematic. I would suggest removing these two figures.

Author Response

  1. The authors mentioned Figure S2 and Figure S3a earlier than Figure S1. Please revise their order.

Response:

In section 2.1, we put this sentence “The contrast sample of Si/rGO-a was self-assembled as the route illustrated in Fig. S1, which has the same amount of GO but just by once wrapped process” in the front of the paragraph. After modification, the order of the pictures is Fig. S1, Fig. S2, Fig. S3.

  1. In Figure 4d, the authors need to add “this work” inside the figure.

Response:

In Figure 4d, this work is marked with red stars and illustrated with text.

  1. In Figure 5e, the linear relationship between Log(current) and Log(scanning rate) is poor. Usually, most literature can have an excellent linear result with a R2 = 0.99-1. However, this work did not have good linearity, which makes the b value and capacitive/diffusive capacity contribution questionable and problematic. I would suggest removing these two figures.

Response:

Thank you for your advice, figure 5e can be used as a reference. In articles (A: 10.1039/c3ee44164d )and (B: org/10.1039/C3EE44164D), the linear relationship between Log(current) and Log(scanning rate) is poor, but it also can as a reference.

(A) V. Augustyn, P. Simon, B. Dunn, Pseudocapacitive oxide materials for high-rate electrochemical energy storage, Energy Environ. Sci. 7 (2014) 1597–1614

(B)Gao, J. Tang, X. Yu, S. Tang, K. Ozawa, T. Sasaki, L.-C. Qin, In situ synthesis of MOF-derived carbon shells for silicon anode with improved lithium-ion storage, Nano Energy 70 (2020) 104444

Reviewer 2 Report

Comments and Suggestions for Authors

Minor revision

Comments

 The authors designed and prepared Si/rGO-b with sandwiched structure, combining ammonia modified silicon (m-Si) nanoparticles (NP) with graphene oxide (GO). This Si/rGO-b composite exhibits excellent cycling performance (Operated stably for more than 800 cycles at high capacity retention rate of 82.4%) and superior rate capability (300 mA h/g at 5 A/g). However, some issues remain in this manuscript. Thus, this manuscript is suitable for publication in the Molecules after minor revision.

 1. The manuscript should address whether the capacities of Si/rGO-b include those of rGO. Furthermore, it should discuss the specific capacities and capacity contributions of rGO under the same voltage window and current density as Si/rGO-b.

 2. Providing dQ/dV differential capacity curves derived from the charge/discharge curves, coupled with CV curves of Si/rGO-b, would enhance understanding of the charge/discharge curve alterations in Si/rGO-b, Si/rGO-a, and Si.

 3. Additional measurements are necessary to comprehensively investigate the stability and changes of active materials (Si/rGO-b, Si/rGO-a, and Si) during lithiation/delithiation processes. Particularly, attention should be given to electrode stability in the electrolyte after cycling.

 4. The manuscript should undergo formatting and language improvements to ensure adherence to standardized and internationally accepted conventions.

 5. The following references are suggested to cite to enhance the readability: https://doi.org/10.1016/j.cej.2021.133234 and https://doi.org/10.1002/smll.202310373

Comments on the Quality of English Language

The manuscript should undergo formatting and language improvements to ensure adherence to standardized and internationally accepted conventions

Author Response

  1. The manuscript should address whether the capacities of Si/rGO-b include those of rGO. Furthermore, it should discuss the specific capacities and capacity contributions of rGO under the same voltage window and current density as Si/rGO-b.

Response:

The capacities of Si/rGO-b include those of rGO. The following figure shows the cycle curve of rGO. The prepared composite Si/rGO-b removes Si by reacting with lye to obtain rGO. rGO cycles charge and discharge at 1A/g current density, with a specific capacity of about 60-80 mAh/g.

Capacity of rGO at 1 C.

  1. Providing dQ/dV differential capacity curves derived from the charge/discharge curves, coupled with CV curves of Si/rGO-b, would enhance understanding of the charge/discharge curve alterations in Si/rGO-b, Si/rGO-a, and Si.

Response:

We provided dQ/dV differential capacity curves of the Si/rGO-b electrode at the 1st cycle. The delithiation/lithiation potential is consistent with CV curve.

Differential discharge-capacity/charge-capacity plots of the Si/rGO-b electrode at the 1st cycle.

  1. Additional measurements are necessary to comprehensively investigate the stability and changes of active materials (Si/rGO-b, Si/rGO-a, and Si) during lithiation/delithiation processes. Particularly, attention should be given to electrode stability in the electrolyte after cycling.

Response:

In this article, figure 5a (EIS) can reflect the properties of electrode materials during charging and discharging. The SEM in figure6a-i shows the stability of the after cycling.

  1. The manuscript should undergo formatting and language improvements to ensure adherence to standardized and internationally accepted conventions.

Response:

We have revised the language of the article according to your suggestion

  1. The following references are suggested to cite to enhance the readability: https://doi.org/10.1016/j.cej.2021.133234 and https://doi.org/10.1002/smll.202310373

Response:  

Thanks for your suggestion, we have cited these two articles, numbers 3 and 8 in the manuscript.

Reviewer 3 Report

Comments and Suggestions for Authors

This manuscript reports about sandwiched silicon/reduced graphene oxide composites for lithium-ion batteries. On the one hand, the authors propose an original and simple methodology for the synthesis of such structures, which show good performance, in particular the Si/rGO-b composite exhibited excellent  rate performance and cycle stability, after 800 cycles at 1 A/g, its specific capacity retained  82.4%, up to 511 mAh/g. On the other hand, such characteristics are not the best ones found in the literature as reported by the authors. For example, in [DOI:10.1021/acsaem.9b01778ACS Appl. Energy Mater.2020, 3, 521-531], better performance is demonstrated and cycling stability is shown over 1000 cycles. Also, the proposed material shows a Coulombic efficiency at the first cycle of only 29 %, which is very low for practical applications. Thus, it is difficult to evaluate the prospectivity of such material. The authors need to explain these issues. In addition, one of the three reasons for the stability of the material used, the authors cite that "Secondly, during the electrochemical process, the oxygen on the surface of m-Si forms Li2O with Li+ to enhance the stability of the SEI film which can assist rGO in buffering the volume expansion of Si.", but the authors did not provide evidence supporting this claim. The validity of the electrochemical impedance spectroscopy measurement technique is also in doubt, as the authors do not provide the potentials at which they perform the measurements. Authors should describe in more detail all equipment and analysis techniques used (SEM, TEM, XPS, EIS and etc.).

Author Response

This manuscript reports about sandwiched silicon/reduced graphene oxide composites for lithium-ion batteries. On the one hand, the authors propose an original and simple methodology for the synthesis of such structures, which show good performance, in particular the Si/rGO-b composite exhibited excellent rate performance and cycle stability, after 800 cycles at 1 A/g, its specific capacity retained 82.4%, up to 511 mAh/g. On the other hand, such characteristics are not the best ones found in the literature as reported by the authors. For example, in [DOI:10.1021/acsaem.9b01778ACS Appl. Energy Mater.2020, 3, 521-531], better performance is demonstrated and cycling stability is shown over 1000 cycles. Also, the proposed material shows a Coulombic efficiency at the first cycle of only 29 %, which is very low for practical applications. Thus, it is difficult to evaluate the prospectivity of such material.

Response:

In the mentioned article, the composite material synthesized by the author has excellent electrochemical performance. However, the author used poly(diallyldimethylammonium chloride) to modify silicon nanoparticles, which had a high viscosity and was easy to cause agglomeration of silicon nanoparticles during the experiment. It is not suitable for large-scale production. The ammonia we use does not cause the agglomeration of silicon nanoparticles. We are also trying hard to improving the initial coulomb efficiency of composites for early industrial applications.

The authors need to explain these issues. In addition, one of the three reasons for the stability of the material used, the authors cite that "Secondly, during the electrochemical process, the oxygen on the surface of m-Si forms Li2O with Li+ to enhance the stability of the SEI film which can assist rGO in buffering the volume expansion of Si.", but the authors did not provide evidence supporting this claim.

Response:

In the articles of [DOI:10.1021/acsaem.9b01778] and [10.1016/j.carbon.2011.01.002] that reported the residual oxygen-containing functional groups of rGO and the SiOx layer on the surface of SiNPs can react with Li+ to form SEI

(1) Wasalathilake K C , Hapuarachchi S N S , Zhao Y ,et al.Unveiling the Working

Mechanism of Graphene Bubble Film/Silicon Composite Anodes in Li-Ion Batteries: From Experiment to Modeling[J]. ACS Appl. Energy Mater.2020, 3, 521-531

(2)Xiang, H.; Zhang, K.; Ji, G.; Lee, J. Y.; Zou, C.; Chen, X.; Wu, J. Graphene/nanosized silicon composites for lithium battery anodes with improved cycling stability. Carbon 2011, 49 (5), 1787−1796

The validity of the electrochemical impedance spectroscopy measurement technique is also in doubt, as the authors do not provide the potentials at which they perform the measurements.

Response:

The voltage data have been recorded during our test. The voltages of electrode materials Si/rGO-b, Si/rGO-a and Si are 1.16399, 1.21473 and 1.07582V, respectively。

Authors should describe in more detail all equipment and analysis techniques used (SEM, TEM, XPS, EIS and etc.).

Response:

We have described in more detail part of equipment and analysis techniques according to your suggestion.